# Hemispheric Asymmetry of the Hand Motor Representations in Patients with Highly Malignant Brain Tumors: Implications for Surgery and Clinical Practice

**DOI:** 10.3390/brainsci12101274

**Published:** 2022-09-21

**Authors:** Elisa Cargnelutti, Giada Pauletto, Tamara Ius, Lorenzo Verriello, Marta Maieron, Miran Skrap, Barbara Tomasino

**Affiliations:** 1Scientific Institute, IRCCS E. Medea, Dipartimento/Unità Operativa Pasian di Prato, 33037 Udine, Italy; 2SOC Neurologia, Azienda Sanitaria Universitaria Friuli Centrale ASU FC, 33100 Udine, Italy; 3SOC Neurochirurgia, Azienda Sanitaria Universitaria Friuli Centrale ASU FC, 33100 Udine, Italy; 4Operative Unit of Neurosurgery, Department of NESMOS, Sapienza University of Rome, 00185 Rome, Italy; 5SOC Fisica Sanitaria, Azienda Sanitaria Universitaria Friuli Centrale ASU FC, 33100 Udine, Italy

**Keywords:** hand clenching, functional asymmetry, sensorimotor cortex, highly malignant brain tumors, plasticity, control-group selection

## Abstract

We addressed both brain pre-surgical functional and neurophysiological aspects of the hand representation in 18 right-handed patients harboring a highly malignant brain tumor in the sensorimotor (SM) cortex (10 in the left hemisphere, LH, and 8 in the right hemisphere, RH) and 10 healthy controls, who performed an fMRI hand-clenching task with both hands alternatively. We extracted the main ROI in the SM cortex and compared ROI values and volumes between hemispheres and groups, in addition to their motor neurophysiological measures. Hemispheric asymmetry in the fMRI signal was observed for healthy controls, namely higher signal for the left-hand movements, but not for either patients’ groups. ROI values, although altered in patients vs. controls, did not differ significantly between groups. ROI volumes associated with right-hand movement were lower for both patients’ groups vs. controls, and those associated with left-hand movement were lower in the RH group vs. all groups. These results are relevant to interpret potential preoperative plasticity and make inferences about postoperative plasticity and can be integrated in the surgical planning to increase surgery success and postoperative prognosis and quality of life.

## 1. Introduction

Highly malignant brain tumors, including glioblastomas and metastases, are characterized by a faster and aggressive growth with respect to low-grade gliomas; this is usually associated with a more rapid onset of neurological symptoms and cognitive deficiencies, which are normally more invalidating [1]. Indeed, malignant tumor growth does not leave time to the brain to successfully reorganize in order to cope with invasion of functional tissue; hence, compensation was observed to occur more substantially following surgery [2,3]. This is observed more likely in the case of lesions affecting brain areas reported to have a low plasticity potential [4,5]. These areas include the sensorimotor (SM) cortex and patients with such lesions tend to develop detectable motor deficits. 

Nevertheless, a certain degree of compensatory neuroplasticity has been reported to occur even in patients with highly malignant brain tumors and even in relation to lesions involving the SM cortex (see reviews [6,7]). Results are poor, given that only a few studies focused exclusively on patients with such tumors. Majos et al. [8], who examined 16 patients with high-grade glioma, reported both pre- and postoperative reorganization involving both the ipsilesional areas and the contralesional homolog cortex. Similarly, Shinoura et al. [9], based on five patients with metastasis, observed both pre- and postoperative reorganization, with greater involvement of the contralesional cortex following surgery. Gibb and coworkers [2] recorded, by intraoperative direct electrical stimulation, ipsilateral reorganization in several sensorimotor sites between two consecutive surgeries (see also [10] for post- vs. preoperative reorganization). In these cases, the role of tumor growth and removal on brain remodeling cannot be clearly dissociated.

One potentially relevant aspect, yet poorly explored from the clinical viewpoint, is that of hemispheric asymmetry associated with hand movement. Hemispheric asymmetry can be defined as the different functional activation across the two hemispheres associated with moving the dominant versus the non-dominant hand. Indeed, in the majority of clinical papers assessing, for instance, the impact of glioma and glioma surgery, the role of handedness and potential hemispheric asymmetry have not been taken into account to discuss findings [11,12,13]; further, when comparing patients to healthy controls, the latter are considered to have a roughly symmetric functional pattern [14,15].

Behaviorally, it was observed that, in right-handers, finger tapping dexterity was lower for the non-dominant (i.e., left) hand, given lower number of taps [16] or longer reaction times [17]. From the functional activation viewpoint, asymmetry was observed, too. In Zhang et al. [18], BOLD signal was higher in the right SM area (associated with left-hand movement) than in the left SM area (right-hand movement) when right-handers performed hand grips at a given frequency. These findings are in line with those reported by Zeng et al. [19], who proposed that the left motor area could simply be an adaptation system entailing fewer neurons to fulfil right-hand movements. 

Nevertheless, results appeared different when taking into account activation volumes. Grabowska et al. [20] observed larger contralateral left-hemisphere (LH) activation in right-handers when moving the right hand and the reverse pattern (i.e., greater contralateral right-hemisphere (RH) activation for the left hand) in left-handers, although asymmetry was less pronounced in the latter. When focusing on the SM area specifically, results were contradicting [18,21]. The non-dominant hand was also observed to determine a greater recruitment of the ipsilateral hemisphere [17,20,22], suggesting the needed recruitment of additional and even homolog regions to accomplish the movement with this hand. This was proposed to be more pronounced in right- than in left-handers [22], as the latter have a less pronounced functional asymmetry related to hand use. 

Furthermore, hemispheric asymmetry is modulated by additional factors, such as task complexity, with asymmetry generally observed to decrease with increasing complexity, because regions other than the primary SM areas are more strongly involved [20,22] (but see [23]). Hence, asymmetry is likely to be maximized by simple tasks. 

Given that, when dealing with patients, it is not possible to determine their previous level of motor functional asymmetry, understanding the pattern of activation in physiological conditions is crucial. A proper selection of the control group might serve this purpose. Accounting for the role of task complexity, we assessed whether asymmetry, usually detected by finger tapping tasks, can also result from execution of easier tasks, such as hand clenching movements. The latter can be sometimes preferred in the clinical practice, as it can be more easily performed by patients who have compromised sensorimotor and/or cognitive processes (e.g., with difficulty in coordinating movements or in understanding which fingers they have to move). 

Therefore, with this study we aimed to investigate potential changes in functional asymmetry as the result of highly malignant tumor growth in either left or right hemispheres. In order to be able to interpret the results correctly, we first had to investigate the pattern in healthy control subjects. Hence, detailed aims were: (i) to explore whether functional hemispheric asymmetry can be observed in normal physiological conditions and during execution of a hand clenching task and then to explore the suitability of this task to detect functional brain differences associated with movement of the dominant vs. the non-dominant hand; (ii) to observe whether such asymmetry was altered in patients with brain malignancies and whether potential differences could be detected across patients with tumors in the left or in the right hemisphere and in relation to hand ipsilateral or contralateral to the affected hemisphere. Indeed, hand impairment and preference following brain damage have been reported to be influenced by which hemisphere is affected [24]; (iii) to investigate whether changes in the SM cortex activation could be interpreted in terms of plasticity; (iv) holding these results, to delineate a few guidelines for interpreting clinical data.

The final aim was that of better understanding whether a simple motor tasks, such as hand clenching, can be successfully used in the clinical assessment and how results associated with either hand have to be interpreted in terms of potential brain functional impairment or compensation.

## 2. Materials and Methods

### 2.1. Participants

We retrospectively identified two group of patients harboring a highly malignant brain tumor (glioblastoma or metastasis) either in the left (LH group) or in the right (RH group) SM cortex and who did not have additional neurological or psychiatric disorders. The LH group included 10 patients (six females) and the RH group eight patients (four females). All the patients were right-handed (as tested by the Edinburgh Inventory [25], which attributed handedness values in the range −100–+100, namely from perfect left-handedness to perfect right-handedness, based on the reported use of either left or right limb to perform 12 common motor actions). The two groups did not differ significantly in age (LH group (years): *M* = 56.90, *SD* = 12.37; RH group (years): *M* = 48.63, *SD* = 8.58; *U* = 22.50, *p* = 0.12) or lesion volume (LH group (cm^3^): *M* = 37.18, *SD* = 29.20; RH group (cm^3^): *M* = 31.84, *SD* = 14.35; *U* = 38.00, *p* = 0.86). 

Patients underwent a neuroimaging (see below) and a neurologic assessment. The latter included an objective evaluation and measurement of motor-evoked potentials (MEPs). MEPs were induced by MagPro stimulator (MagVenture) and data acquired using the Dantec Keypoint multi(6)-channel electromyography system. While stimulation facilitates identification of functional tissue, motor evoked potentials (MEPs) can be utilized to monitor motor function during all of the tumor resection phases. This may be performed transcranially or by direct cortical stimulation via a grid or strip electrode [26]. In our study population, cortical magnetic stimulation was bilaterally applied to the motor cortex and the elicited action potentials were recorded via a surface electrode placed at the first dorsal interosseous muscle of either hand. The response was measured as both the time (milliseconds) required for the electrical impulse to travel from the stimulation site to an electrode placed at the recording site and the amplitude (millivolts) of the evoked response.

MEPs were also measured intraoperatively (together with sensory-evoked potentials). All our surgical procedures (both awake and under general anesthesia) were conducted under cortical and subcortical direct electrical stimulation. At the cortical level, a maximum of 4 mA of current intensity was sufficient, whereas, in subcortical mapping, we usually started at 6 or 8 mA. Electroencephalography and electrocorticography recordings were also performed during awake surgeries to control the occurrence of after-discharge or intraoperative focal short-lasting seizures [27]. 

Patients’ demographic and clinical information is detailed in Table 1 (see also Table 2).

The control group included 10 healthy right-handed volunteers (six females), who had no history of neurological or psychiatric disorders. Their mean age (47.50, *SD* = 8.00) did not significantly differ from that of patients in either the LH group (*U* = 26.00, *p* = 0.07) or RH group (*U* = 39.00, *p* = 0.93). Further, the groups were approximately gender balanced (non-significant Fisher’s exact probability test: 3.62, *p* = 1.00). 

All the participants signed an informed consent to participate in the research project, in agreement with the Declaration of Helsinki. The study was approved by the Ethics Committee of the IRCCS E. Medea (protocol No. 39/08-CE). 

### 2.2. fMRI Data Acquisition

Participants were asked to perform the motor task while lying in the scanner with arms relaxed along the body. They were instructed to open and close either hand according to the visual cue (i.e., arrow) appearing on the screen together with the instruction “Move”. The movement was performed at a self-paced rate—in order to reduce task demands and effort—and until the instruction changed to “Still”. Each block lasted 15 s for a total of eight task blocks (i.e., four for the right hand and four for the left hand) interleaved with nine rest blocks.

Images were acquired on a Philips Achieva 3-T (Best, Netherlands) whole-body scanner using a SENSE-Head-8 channel head coil and a custom-built head restrainer to minimize head movements. Functional images were obtained using a single-shot gradient echo, echoplanar imaging (EPI) sequence. EPI volumes (*N* = 102) contained 34 contiguous axial slices (repetition time, TR = 2500 msec, echo-time, TE = 35 msec, field-of-view, FOV = 230 × 230 mm^2^, matrix: 128 × 128, voxel size: 1.797 × 1.797 × 3 mm^3^, 90° flip angle).

Stimulus sequence was designed and synchronized with the MR scanner by the Presentation software (Version 9.9, Neurobehavioral Systems Inc., Berkeley, CA, USA) and were displayed by the VisuaStim Goggles system (Resonance Technology Inc., Northridge, CA, USA). 

### 2.3. fMRI Data Processing

Imaging analyses were performed using MATLAB r2018a (The Mathworks Inc., Natick, MA, USA) and SPM12 (Statistical Parametric Mapping software, SPM; Wellcome Department of Imaging Neuroscience, London, UK http://www.fil.ion.ucl.ac.uk/spm). 

Image pre-processing included spatial realignment to reference volume, segmentation, and normalization to standard anatomical template. Image voxels were then resampled to 2 × 2 × 2 mm^3^ voxel size and spatially smoothed with a 6-mm FWHM Gaussian kernel.

In first level analysis, we defined the network associated with right-hand moving (RHM), left-hand moving (LHM), and rest conditions for each participant. Hence, we modelled the alternating epochs by a simple boxcar reference vector. A general linear model was applied to each voxel for alternating task and baseline (rest) conditions; the temporal derivatives were modelled by means of reference waveforms corresponding to boxcar functions convolved with a hemodynamic response function [29,30]. Six additional regressors modelled the head movement parameters from the realignment procedure. A design matrix was then built by defining linear contrasts between the two task and rest conditions, which resulted in a *t*-statistics for each voxel. Low-frequency signal drifts were filtered using a cut-off period of 128 s. T-statistics were then transformed into z-statistics constituting statistical parametric maps (SPM{*Z*}) of differences between task and baseline conditions.

For second-level random effects analysis, contrast images from each participant were entered into a one-sample *t*-test to create a SPM{*T*} for each contrast at the group level. In particular, we focused on the following SPM{*T*}: RHM > rest and LHM > rest. 

Resulting contrast maps were corrected for multiple comparisons at the cluster level (i.e., family-wise error (FWE) correction, *p* < 0.05), with a height threshold of *p* < 0.001, uncorrected, at the voxel level. Anatomical label to the resultant functional clusters was performed using the SPM Anatomy toolbox [31]. The macro-anatomical localization of the functional activation and, when assignable, the cytoarchitectonic localization, too, were then provided. 

### 2.4. ROI Analysis

Region of interest (ROI) analyses were conducted using MarsBar [32], which runs on SPM8. Statistical analyses on the extracted values related to ROI differences were performed by IBM SPSS Statistics, version 21 (www.ibm.com). Analysis focused on the contralateral SM cortex, where activation was confirmed to be the strongest in all the control subjects. 

We extracted the correspondent ROI from first-level analysis results, namely from the RHM > rest and LHM > rest contrasts for each participant. We then saved the cluster of activation with the local maxima and extracted the correspondent values for the two ROIs, in the left and right hemispheres, respectively. For each ROI, the 102 extracted values corresponded with the time-course mean signal from all the voxels within each ROI. We averaged the values across all the blocks associated with either RHM or LHM within each ROI, to obtain a mean value for each subject for the left ROI (associated with RHM) and the right ROI (associated with LHM). 

ROIs were then inspected in detail by MRIcron software (https://www.nitrc.org), in order to achieve more detailed information concerning ROI volume and proportion of activated SM cortex. ROI volume was expressed in terms of number of non-zero, activated, voxels, whereas proportion of activated SM cortex as the ratio between number of non-zero voxels activated in the SM cortex specifically and total number of voxels constituting the SM cortex.

We then ran a between-group comparison for ROI signal strength values (hereafter, ROI values), non-zero ROI voxels (hereafter, ROI volumes), and proportion of activated SM cortex (hereafter, SM activation) associated with each hand. Further, we checked for hemispheric asymmetry in the group of healthy controls and tested whether such asymmetry was altered in the patients’ groups. 

Due to small and unbalanced sample sizes and inhomogeneous variance, we ran non-parametric comparisons (i.e., Kruskal–Wallis and Mann–Whitney tests).

## 3. Results

### 3.1. Descriptive Results

The ROI was centered in the contralateral SM cortex (i.e., pre- and post-central gyrus) in all the healthy controls (10/10) with regards to both RHM and LHM conditions. In neither condition there was an activation of the ipsilateral cortex. The ROI extended to neighboring areas (e.g., superior frontal gyrus, inferior parietal cortex), but their activation varied across subjects and did not reach full overlap among them.

Concerning patients, in the LH group, the ROI was centered in the contralateral SM cortex in 9/10 patients for both RHM and LHM conditions. For one patient (p7), the ROI could not be identified at all in either condition (i.e., hemisphere). In the RH group, contralateral SM cortex activation occurred in 8/8 patients for RHM (although, in 1/8 patients, it involved the postcentral gyrus only) and in 7/8 patients for LHM (although, in 2/8 patients, it involved the precentral gyrus only). ROI overlap for each condition and group is depicted in Figure 1.

Furthermore, a few patients (three in the LH group and two in the RH group), but none of the healthy controls, had a small ipsilateral SM activation in the healthy hemisphere associated with movement of the affected hand. These ROIs varied largely in values and volumes (LH group: p1: *M*_value_ = 99.72, *M*_volume_ = 9; p3: *M*_value_ = 109.06, *M*_volume_ = 29; p7: *M*_value_ = 214.91, *M*_volume_ = 12; RH group: p7: *M*_value_ = 138.96, *M*_volume_ = 5; *M*_value_ = 137.77, *M*_volume_ = 113).

### 3.2. Hemispheric Effects: Within-Group Comparison between RHM and LHM Conditions

Table 2 reports detailed ROI information. In healthy controls, ROI values significantly differ between the two hemispheres, with them being greater for the ROI associated with LHM vs. RHM (*z* = −2.60, *p* < 0.01). ROI volume, expressed as number of non-zero (i.e., activated) voxels, did not significantly differ between the conditions (*z* = −1.38, *p* = 0.17) and the same occurred for SM activation (*z* = −0.97, *p* = 0.33).

For patients in the LH group, differences between LHM and RHM conditions were not significant for any assessed parameters (ROI values: *z* = −1.00, *p* = 0.31; ROI volumes: *z* = −1.36, *p* = 0.17; SM activation: *z* = −1.24, *p* = 0.21). The same was observed for the RH group (ROI values: *z* = −0.98, *p* = 0.33; SM activation: *z* = −1.68, *p* = 0.09), although difference in ROI volumes approached significance due to lower volume associated with the LHM condition (*z* = −1.82, *p* = 0.06). See Table 3.

### 3.3. Between-Group Comparisons

#### 3.3.1. Separate Comparisons for RHM and LHM Conditions between Groups

Non-parametric between-group comparisons showed that differences in ROI values did not achieve statistical significance for either RHM (*χ*^2^ = 2.08, *p* = 0.35) or LHM (*χ*^2^ = 4.46, *p* = 0.10) conditions. Concerning ROI volumes, instead, we observed a significant difference for both RHM (*χ*^2^ = 12.53, *p* < 0.01) and LHM (*χ*^2^ = 15.17, *p* < 0.01) conditions. Between-group difference was also significant regarding SM activation for both RHM (*χ*^2^ = 11.79, *p* < 0.01) and LHM (*χ*^2^ = 15.30, *p* < 0.001) conditions. 

Post-hoc pairwise comparisons showed that, for the RHM condition, healthy controls had both significantly larger volume and SM activation than patients in both the LH group (respectively: *U* = 5.00, *z* = −3.27, *p* < 0.01; *U* = 7.00, *z* = −3.10, *p* < 0.01) and RH group (respectively: *U* = 10.00, *z* = −2.67, *p* < 0.01; *U* = 10.00, *z* = −2.67, *p* < 0.01). Concerning the LHM condition, both controls and patients in the LH groups had significantly larger volumes (respectively: *U* = 0.00, *z* = −3.55, *p* < 0.001; *U* = 8.00, *z* = −2.69, *p* < 0.01) and SM activation (respectively: *U* = 0.00, *z* = −3.55, *p* < 0.01; *U* = 10.00, *z* = −2.50, *p* = 0.01) than patients in the RH group.

#### 3.3.2. Differences in Hemispheric Asymmetry between Groups

We ran the same analyses by taking into account the arithmetical difference (delta) in ROI values, volumes, and SM activation between the two conditions (i.e., Δ_LHM-RHM) at the group level. By taking these values, we tested between-group difference in asymmetry, and we found it to approach statistical significance for ROI values (*χ*^2^ = 5.30, *p* = 0.07) and be significant for ROI volume (*χ*^2^ = 6.59, *p* = 0.03) and SM activation (*χ*^2^ = 6.77, *p* = 0.03). 

Pairwise comparisons showed that the difference associated with ROI values was significant for the healthy controls’ group vs. RH group (*U* = 16.00, *z* = −2.13, *p* = 0.03); difference associated with ROI volume was significant for the healthy control group vs. LH group (*U* = 20.00, *z* = −2.04, *p* = 0.04) and for the RH vs. LH group (*U* = 12.00, *z* = −2.31, *p* = 0.02); finally, difference associated with SM activation was significant for the RH vs. LH group (*U* = 10.00, *z* = −2.50, *p* = 0.01).

## 4. Discussion

With the current study, we aimed to investigate changes occurring in functional activations associated with movement of the dominant versus the non-dominant hand in patients harboring a highly malignant brain tumor in the SM cortex. From the literature on healthy individuals, it is known that activation of the contralateral SM cortex, as elicited by tasks such as finger tapping, is characterized by a functional asymmetry in the ROI signal strength values, which were observed to be higher when tapping was performed by the dominant versus. non-dominant hand, meaning right versus left hand in right-handed subjects [18,19]. No univocal results were instead reported concerning ROI volumes [18,20,21].

In our routine clinical assessment, we tested hand motor functions by a hand clenching task performed at a self-paced rate, given that this task is less demanding to patients from both the motor and the cognitive viewpoints, and it addresses the hand motor function more purely. As result, in order to interpret clinical data, we investigated, first, whether hemispheric functional asymmetry can be recorded when performing this easier task. Second, we aimed to investigate whether such potential asymmetry could be found in patients harboring a highly malignant tumor in the left or in the right hemisphere and whether these findings could tell us something about tumor effects on the SM cortex functioning.

Our findings indicate that, in physiological conditions (i.e., healthy controls), functional asymmetry is recorded during execution of hand clenching tasks as well; indeed, ROI values were higher when performing movements with the non-dominant hand, namely in right SM cortex associated with the LHM condition. These results mirrored findings on functional asymmetry from previous studies [18,19] and did not show any involvement of the ipsilateral SM cortex, in agreement with studies reporting this activation for more complex hand motor tasks [20,22]. Concerning differences in ROI volumes, regarding which previous studies reported contradicting findings [18,20], we did not find any significant differences across the two hemispheres/conditions either in ROI volumes or in proportion of activated SM cortex. These finding suggest that potential involvement of the ipsilateral SM cortex in patients is likely to reflect compensatory processes. 

When taking patients into account, we did not observe the above-described pattern of asymmetry in ROI values, as these were more comparable between the two conditions. For the LH group, we did not note any significant difference, whereas, in the RH group, there was a tendency towards significantly lower ROI volume associated with LHM, meaning in the affected hemisphere. 

After having investigated within-group phenomena, we ran between-group comparisons. When analyzing the two conditions separately, no significant differences emerged between groups in ROI values, but, when taking into account delta values (i.e., Δ_LHM-RHM), a significant difference was recorded in patients in the RH group versus healthy controls. This was due to an inverted asymmetry pattern, meaning lower ROI values in the affected hemisphere associated with LHM vs. RHM. 

Concerning both ROI volumes and proportion of SM cortex activation, we detected significant differences, which did not represent the physiological condition detected in healthy controls. In particular, patients in the LH group had both smaller ROI volume and proportion of SM cortex activation than healthy controls for the RHM condition, hence in the affected hemisphere. On the contrary, patients in the RH group had significantly lower ROI volumes and SM activation for both conditions; nevertheless, this difference was more marked in the affected hemisphere, in which they significantly differ from patients in the LH group, too. The latter had, as well, lower values for both conditions, but those in the healthy hemisphere did not reach statistical significance. Therefore, reduction in whole ROI volume was paralleled by specific reduction in the proportion of activated SM cortex.

Taken together, our findings indicate that the presence of a highly malignant tumor in the SM cortex determines relevant changes with respect to the normal activation conditions and in a way that requires evaluating which is the affected hand and whether it is the dominant one. Unfortunately, this evaluation does not typically occur in the clinical practice (see [11,12,13]). We observe significant changes to take place in the hemisphere harboring the lesion but, to a certain extent, in the healthy hemisphere, too. This finding is in agreement with a recent study reporting the long-range effects of highly malignant lesions, capable of affecting the contralesional hemisphere, as well [33]. This phenomenon was more marked for patients in the RH group, for whom ROI overlap across patients was low for both conditions (see Figure 1).

Unfortunately, the number of patients was too small to enable analyses based on their clinical status. Nevertheless, by inspecting Table 1 and Table 2, it is possible to notice that patients in the RH group were, overall, more affected clinically. ROI data, then, are likely to reflect the clinical patients’ conditions. Discrepancy between the two patients’ groups can be attributed to chance and, probably, with different (and larger) patients’ samples, it would not be detected. As groups were matched in lesion volumes, then differences could not be attributed to this parameter. Nevertheless, we could not exclude that patients in the RH group had worse clinical and ROI data because of a greater vulnerability of the non-dominant hand. Results from healthy controls, namely higher ROI values for the LHM condition, seem to suggest a higher effort associated with movement of the non-dominant hand. Hence, given that motor functions are highly localized and plasticity in associated brain areas is more limited [5], it is possible that a lesion in this area may have a more detrimental effect for the more susceptible hand. 

For what concerns potential plasticity, which is a highly debated topic in relation to both highly malignant lesions and motor functions [6,7], our data cannot support it. Our general observation is that functional impairment in the SM cortex is likely to be associated with a poorer clinical status. Further, we observed homolog SM cortex activations in a few patients only, but this could not be associated with a better clinical outcome for these patients. As reported in other studies [2,10], functional rearrangement could take place more consistently in the post-operative period, as a longer time window is probably necessary to detect significant rearrangement. Importantly, patients with highly malignant lesions, as those included in our study, are frequently elderly and older age is another limitation to plasticity.

The current study suffers from a few limitations. An important limitation is represented by small patients’ samples. This prevented, for instance, to address additional aspects relating brain function to clinical data. In future studies, it will be advisable to explore more deeply the relation between SM cortex functional integrity, potential plasticity, and clinical status. Creation of a patients’ database, possibly also in the perspective of a multicentric study, is advisable.

Another related limitation is the lack of a group of left-handed participants, which prevented generalization of findings. Nevertheless, we can hypothesize generalization in terms of hand dominance: A stronger contralateral activation in the SM cortex is likely to be found in the left hemisphere of left handers when moving their right, non-dominant hand, although hemispheric asymmetry is probably less marked in this population [16,20,34]. 

Concerning data analyses, we cannot exclude SM cortex activation to be influenced by mass effect and potential neurovascular uncoupling [35,36], even though several studies reported reliability of functional SM activations even in the presence of high-grade lesions [13,37,38].

## 5. Conclusions

Findings from the present study may have important implications for the clinical practice, in that they propose important tips to be used in the clinical assessment of patients with highly malignant tumors in the SM cortex. The study shows that even the more patient-friendly hand clenching task enables to detect, in physiological conditions, hemispheric differences. Therefore, this task can be used successfully to test residual motor functions in patients, without posing further demands to them; in this way, detected functional activations are more likely to merely reflect movement per se. 

In order to reliably detect motor activation and related preserved functional asymmetry in patients, it is important to pay attention to several variables and, first, to assessed hand: Is it the dominant hand? Does it correspond to the affected hemisphere? As it is not possible to determine previous level of functional asymmetry in patients, understanding which is the general pattern of activation in physiological conditions is crucial to interpret clinical data. Second, in line with this and in the perspective of neurosurgery, determining the degree of SM cortex impairment by performing a comparison between SM activation elicited by affected versus unaffected hand could be improper, as asymmetry is a physiological condition. Rather, it should be compared to the corresponding hand of a properly selected control group, which should be selected by taking into account additional variables, such as handedness, age, and gender [39,40].

## Figures and Tables

**Figure 1 brainsci-12-01274-f001:**
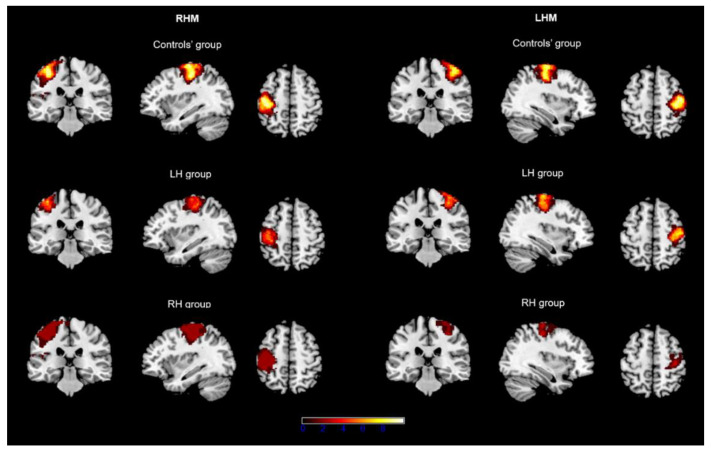
Rendered representation of ROI overlap in each group associated with right-hand (RHM) and left-hand (LHM) movement. Note. Images are reported in neurological convention. Color bar represents overlap intensity (i.e., number of participants for whom ROIs overlapped in a given area).

**Table 1 brainsci-12-01274-t001:** Patients’ demographic and clinical information.

	Gender	Age	Edu	Diagnosis	Vol (cm^3^)	ONA Pre	ONA Post	MEP Pre	MEP Intra	Epil Pre	Epil Post (Engel Class)
**LH**											
p1	M	57		Meta	0.52	R hand myasthenia (4/5)	R superior limb myasthenia	Patho	Normal	Yes	No (Ia)
p2	F	59		Meta	59.60	NAD	NAD	Normal	Normal	Yes	No (Ia)
p3	F	64		GBM	23.15	N/A	N/A	N/A	N/A	N/A	N/A
p4	M	65		GBM	22.86	NAD	R hemiparesis (2/5)	Patho	Normal	Yes	No (Ia)
p5	M	70		GBM	12.56	NAD	NAD	Normal	Normal	Yes	Yes (III)
p6	F	38		GBM	61.79	NAD	R hemiparesis (3/5)	Patho	Normal	Yes	No (Ia)
p7	M	37		GBM	96.20	N/A	N/A	N/A	N/A	N/A	N/A
p8	F	52		GBM	41.64	R hemisome myasthenia (3/5)	R hemisome myasthenia (2/5)	Patho	Patho	Yes	Yes (III)
p9	F	74		Meta	42.50	NAD	R hemisome myasthenia (2/5)	Normal	Normal	No	No (Ia)
p10	F	53		GBM	11.03	N/A	N/A	N/A	N/A	N/A	N/A
**RH**											
p1	M	44		Meta	29.25	N/A	N/A	N/A	N/A	N/A	N/A
p2	M	37		GBM	23.82	NAD	NAD	Normal	Normal	Yes	No (Ia)
p3	F	49		GBM	30.98	L hemiparesis (3/5)	L hemiparesis (3/5)	Patho	Patho	Yes	Yes (IV)
p4	F	40		GBM	48.16	NAD	Recovering L hemiparesis (4/5)	Normal	Normal	Yes	Yes (IV)
p5	F	54		GBM	52.21	N/A	N/A	N/A	N/A	N/A	N/A
p6	M	65		GBM	39.15	L superior limb myasthenia (3/5)	L hemiparesis (2/5)	Patho	Patho	Yes	Yes (IV)
p7	F	48		GBM	8.31	L superior limb myasthenia (4/5)	L hemiparesis (3/5 superior limb, 1/5 inferior limb)	Patho	Normal	Yes	Yes (III)
p8	M	53		GBM	22.83	L superior limb myasthenia (3/5)	L hemiparesis (2/5)	Patho	Patho	Yes	Yes (Ib)

Note. Edu: years of education; GBM: glioblastoma; intra: intraoperative; L: left; LH: patients with left-hemisphere lesion; MEP: motor-evoked potentials; Meta: metastasis; N/A: not available; NAD: no abnormality detected; ONA: objective neurological assessment; p: patient’s number; Patho: pathological; pre: preoperative; post: postoperative; R: right; RH: patients with right-hemisphere lesion; Vol: lesion volume. Engel class refers to classification in the Engel Epilepsy Surgery Outcome Scale [28], where class Ia corresponds to complete absence of seizure since surgery.

**Table 2 brainsci-12-01274-t002:** Number of patients with affected clinical parameters in each patients’ group.

	Preoperative MEP	Intraoperative MEP	Preoperative ONA	Postoperative ONA	Engel Class Other than Ia
LH group (*n* = 10)	4	1	2	4	2
RH group (*n* = 8)	4	3	4	5	5

Note. MEP = motor evoked potentials; ONE = objective neurological assessment. Engel class refers to classification in the Engel Epilepsy Surgery Outcome Scale [28], where class Ia corresponds to complete absence of seizure since surgery.

**Table 3 brainsci-12-01274-t003:** ROI parameters associated with hand clenching in the three study groups.

	RHM			LHM					
	ROI Values	ROI Volumes	SM Activation (%)	ROI Values	ROI Volumes	SM Activation (%)	Δ_ROI Values	Δ_ROI Volumes	Δ_SM Activation (%)
Control group	136.02 (4.85)	11,656.60 (4874.10)	16.41 (6.13)	145.63 (9.11)	9414.70 (2933.90)	14.63 (6.13)	9.53 (3.44)	−2241.90 (4,441.75)	−1.77 (5.85)
LH group	132.13 (13.17)	4496.89 (1918.73)	7.22 (2.86)	136.55 (8.63)	6725.56 (3293.93)	10.36 (5.06)	4.42 (3.63)	2228.67 (4040.98)	3.14 (6.13)
RH group	140.23 (13.90)	5008.50 (3596.33)	7.64 (5.04)	134.79 (13.26)	1895.13 (1705.91)	3.01 (2.83)	−5.44 (3.85)	−3113.38 (4340.30)	−4.63 (6.39)

## Data Availability

Data will be provided upon direct request to the corresponding author.

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
