# Peer review of "Hemispheric Asymmetry of the Hand Motor Representations in Patients with Highly Malignant Brain Tumors: Implications for Surgery and Clinical Practice"

_brainsci, 2022, doi:10.3390/brainsci12101274_

Round 1

Reviewer 1 Report

Interesting article of relevance to understanding normal functional activation of sensorimotor cortex as well as changes resulting from harboring tumors in this location.  

Author Response

We thank the Reviewer for having appreciated our manuscript. We checked English grammar and syntax. We upload the revised version of the ms, with the changes requested by the other reviewers.

Reviewer 2 Report

The work submitted by the authors examines the brain activation associated with the hand movement in patients with malignant tumour in the sensorimotor cortex. The study design and MRI analyses are appropriate. Some questions:

Line 17: I suggest use sensorimotor cortex (SM) for consistency, also because it is likely that the tumour area may be well involve S1.

Line 52: typo in cannot’t

Table 1: Can the authors briefly write in the Methods section how MEPs were obtained?

Line 158: Any head motion correction involved in SPM?

Line 163: ‘anatomical template’ is better than ‘EPI template’; as the MNI standard image is not a functional image, but a 3D T1 image.

Limitation: I appreciate the small number of patients in LH/RH. I suggest the authors continue this work with more patients involved, e.g. creation of a database will be useful for neurologists or neurosurgeons.

Author Response

We thank the Reviewer for positive comments on our work and for suggestions aimed at improving it. Accordingly, we amended the alerted typos and put on evidence the sentence regarding motion correction in SPM (“Six additional regressors modelled the head movement parameters from the realignment procedure”). In the method section, we explained how MEPs were recorded. Following the Reviewer’s suggestion, we stressed the need of creating a patients’ database, with a higher number of patients, to be used in future studies.

All these changes are marked in the text, together with those requested by the other reviewers.

Reviewer 3 Report

Thank you for the opportunity to review this manuscript. In attachament you can find my recommnedations. 

Author Response

We thank the Reviewer for positive comments on our work and for suggestions aimed at improving it. Accordingly, concerning comment at line 117, we added a few information regarding the questionnaire we adopted to test for hand laterality: “…. as tested by the Edinburgh Inventory, [25], which attributed handedness values in the range -100-+100, namely from perfect left-handedness to perfect right-handedness, based on the reported use of either left or right limb to perform 12 common motor actions.”

Regarding comments referred to Discussion and closing paragraphs, we better commented our results in reference to previous literature and we shifted the limitation section under the Discussion section, to then better detail Conclusions. We improved the method section, too.

Finally, we checked the References section and amended it so that all the references could meet the journal requirements.

All these changes are marked in the text, together with those requested by the other reviewers.